# Effect of Balance Training in Sitting Position Using Visual Feedback on Balance and Gait Ability in Chronic Stroke Patients

**DOI:** 10.3390/jcm12134383

**Published:** 2023-06-29

**Authors:** Sang-Seok Yeo, Dong-Kyun Koo, Seong-Young Ko, Seo-Yoon Park

**Affiliations:** 1Department of Physical Therapy, College of Health Sciences, Dankook University, 119 Dandae-ro, Dongnam-gu, Cheonan-si 31116, Republic of Korea; eangbul@hanmail.net; 2Department of Public Health Sciences, Graduate School, Dankook University, 119 Dandae-ro, Dongnam-gu, Cheonan-si 31116, Republic of Korea; definikk@gmail.com; 3Department of Physical Therapy, Graduate School of Health and Welfare, Dankook University, 119 Dandae-ro, Dongnam-gu, Cheonan-si 31116, Republic of Korea; 4Department of Physical Therapy, College of Health and Welfare, Woosuk University, 443 Samnye-ro, Samnye-eup, Wanju-gun 55338, Republic of Korea

**Keywords:** balance training, visual feedback, stroke

## Abstract

Chronic stroke often results in balance and gait impairments, significantly impacting patients’ quality of life. The purpose of this study was to investigate whether the combined effect of unstable surface balance training and visual feedback, based on proprioceptive neuromuscular stimulation in patients with chronic stroke, is effective in restoring balance and gait ability. A total of 39 chronic stroke patients were randomly assigned to a visual feedback combined with unstable surface balance training group (VUSBG), an unstable surface balance training group (USBG), or a conventional physical therapy group (CG). This study was conducted with the Trunk Impairment Scale, the Bug Balance Scale, the Timed Get Up and Go Test, and Gait Analysis. VUSBG and USBG improved function and gait (stride length and hip/knee flexion angle), but there was no significant difference in the CG group. Specific results showed that the stride length in the VUSBG improved by 25% (*p* < 0.05), and the hip/knee flexion angle improved by 18% (*p* < 0.05). The post-hoc analysis revealed that VUSBG had a greater impact on the hip/knee flexion angle relative to the other two groups, as well as gait velocity and stride length relative to CG. Visual feedback complex exercise based on the principle of proprioceptive neuromuscular facilitation could be an intervention strategy to improve gait speed, trunk stability, and mobility in chronic stroke patients.

## 1. Introduction

Stroke is a neurological deficit caused by a blood vessel rupture in the brain that leads to bleeding, or due to a blockage in the blood supply to the brain. The rupture or blockage prevents the flow of blood and oxygen to the brain tissue and results in dysfunction related to the location and extent of the injury [1,2,3]. The most typical symptom is hemiplegia of the brain lesion on the contralateral body [4]. These symptoms are characterized by an asymmetrical upright posture due to weight-bearing control and balance loss on the affected side [5]. Moreover, asymmetric posture affects gait patterns, which limit slow and independent gait [6]. Therefore, they tend to experience difficulty in achieving balance when standing or sitting.

Chronic stroke impairs sitting balance and trunk function. Physical balance requires proper control of the center of gravity and movement, as well as the processing of complex sensory and motor inputs [7]. Previous studies have reported the importance of trunk stabilization exercises in stroke patients. It is known that selective trunk stabilization exercises in supine and sitting positions improve gait ability, especially balance and trunk control in stroke patients [8]. Poor postural stability of the trunk post-stroke disrupts the recovery of overall functional performance, so the ability to maintain posture in an unsupported sitting state is required [9]. Impaired sitting balance is caused mostly by muscle weakness, sensory deficits, and a tendency to adopt compensatory strategies to avoid threats to balance, resulting in further functional impairment. Therefore, improved sitting balance and trunk function are essential components of rehabilitation after stroke.

Clinical methods for improving balance and gait function in stroke patients include motor dual-task training, computerized postural control training [10,11], sensorimotor training exercise [12], and a method to enhance sensorimotor feedback by activating motor control strategies [13]. Rehabilitation programs maintain a focus on improving balance and movement for greater functional independence. Visual feedback is one of the most widely incorporated external stimuli to improve balance ability in stroke patients and allows them to recognize its position and trajectory in real-time, thus they can take appropriate strategies to stabilize postural control [14]. Balance training is a common approach used to address these impairments and improve a patient’s overall function and quality of life [15]. There are two main types of balance training: Stable and unstable. Stroke patients can benefit from both unstable and stable balance training but for different reasons. Stable balance training can improve confidence, strength, and proprioception, while unstable balance training can challenge balance and promote neural plasticity [16,17]. The type of training employed should be tailored to the needs and abilities of each individual patient.

Visual feedback is important for stroke patients because it can help them improve their balance and postural control [18]. Stroke frequently causes sensorimotor system impairments, which can impair a patient’s ability to maintain balance and prevent falls. Visual feedback that provides real-time information about a patient’s body position and movement can assist patients in compensating for these impairments by permitting them to adjust their movements and postural control strategies accordingly [19,20]. In addition, research has demonstrated that visual feedback can enhance motor learning and promote neural plasticity, which is the brain’s ability to reorganize and form new connections in response to injury [21,22]. This suggests that visual feedback may not only help stroke patients improve their balance in the short term but may also contribute to their long-term rehabilitation.

In recent years, it has been reported that stroke patients could obtain a functional improvement in postural control by integrating visual feedback into well-structured, individualized physiotherapy. Therefore, it could be effective to use visual feedback in conjunction with individualized sitting balance exercises for stroke patients. Training and exercise with visual feedback on an unstable surface can help stroke patients improve their balance and postural control [23], enhance proprioception [24], and promote neuroplasticity [25]. It is crucial that stroke patients collaborate with a trained healthcare professional to develop an exercise regimen that is tailored to their individual needs and abilities.

As previous studies have shown, balance and gait impairments are prevalent among stroke patients, and these impairments have a substantial impact on their quality of life. However, studies that compare the effectiveness of combined exercise interventions are limited. In this study, we investigated the effects of visual feedback combined with unstable surface balance training based on proprioceptive neuromuscular facilitation, which could be useful in promoting the recovery of balance and gait ability in chronic stroke patients.

## 2. Materials and Methods

### 2.1. Participants

Participants were recruited for this study from SAERONA HOSPITAL rehabilitation center, which specializes in stroke rehabilitation. We assigned the research subjects to groups using the blocked-randomization method and ensured that the single-blind method was used to conceal the group allocation from the study participants (https://www.randomizer.org/, accessed on 10 January 2022) (G-POWER; F tests, ANOVA: Fixed effects, omnibus, one-way; Effect size: 0.5, α-err-prob: 0.05, Power (1-β err-prob): 0.8, Number of groups: 3). These procedures were used to reduce bias and increase the reliability of the results. The present study was a randomized controlled trial involving 45 chronic stroke patients who were randomly allocated to one of three groups: A visual feedback unstable surface balance training group (VUSBG), an unstable surface balance training group (USBG), or a conventional group (CG). The inclusion criteria for participants were as follows: Individuals with a confirmed neurologist diagnosis of stroke, at least 6 months post-diagnosis, the ability to stand and walk independently for at least 10 min over a distance of 6 m, no cognitive or visual impairments, ability to understand simple instructions (Mini-Mental State Examination, MMSE > 24), and ability to discriminate colors. Exclusion criteria included individuals with medical conditions that were unstable, severe joint deformities or contractures, and other neurological conditions that could affect gait or balance. All subjects had no severe aphasia or problems with visual integrity, and all were able to clearly see the visual feedback provided on the computer monitor. All patients were given a comprehensive set of instructions regarding the experiment, agreed to the experimental protocol, and provided written informed consent to participate in the study. All experiments were performed in accordance with relevant guidelines and regulations from the Declaration of Helsinki. The measurements were conducted following the protocol approved by the Institutional Review Board (IRB) of Dankook University (DKU 2021-04-026-002).

### 2.2. Measurements

#### 2.2.1. Trunk Impairment Scale and Berg Balance Scale

Clinical assessment of stroke patients was based on specific tests for evaluating trunk control and balance: The Trunk Impairment Scale (TIS) and the Berg Balance Scale (BBS). TIS and BBS tests have been validated for use in stroke patients and have been used to characterize balance deficits. Test–retest and interobserver reliability for the TIS total score (ICC) were 0.96 and 0.99, respectively. The 95% limits of agreement for the test–retest and inter-examiner measurement error were −2.90, 3.68 and −1.84, 1.84, respectively [26]. The BBS has a high relative reliability with inter-rater reliability estimated at 0.97 (95% CI 0.96 to 0.98) and intra-rater reliability estimated at 0.98 (95% CI 0.97 to 0.99) [27].

#### 2.2.2. Timed up and Go Test

As a fall risk detection test, the TUG test is presented with guidelines. It is used as a more quantitative form of evaluation using various measuring sensors, as well as simple timed measurements [28]. The TUG test consists of several phases: (1) Sit-to-stand in a chair, (2) gait 3 m, (3) turn and return from the 3 m position, and (4) turn to sit in a chair. Participants were instructed to put on their regular shoes, stand up, and walk without a gait aid at a self-paced speed upon the signal ‘go’. To familiarize themselves with the test, each participant performed a practice walk-through before the timed test. Time was recorded using a stopwatch for each test [29]. For the ‘Timed Up & Go’ test, the ICC values for interrater reliability were 0.97 and 0.99. The intra-rater reliability coefficients for the “Timed Up & Go” test were 0.95 and 0.96 [30].

#### 2.2.3. Gait Analysis

Using the inertial measurement units (IMU) sensor, the joint angle and gait parameters were measured. The IMU sensors have been used to analyze human joint movements in laboratories or in living environments [31]. Furthermore, the IMU sensor is beneficial since the system function is validated in the gait study of individuals with impairments using a low-cost and non-invasive technique [32]. In the current study, the IMU sensor was iSen (STT system, San Sebastian, Spain). Data were collected using the iSen software (version 3.8 beta, STT Systems, San Sebastian, Spain). For data collection, six IMU sensors were utilized. Each sensor was attached to the sacrum’s center, the dorsal part of thoracic level 1, and the anterior surfaces of both, the shin (3 cm above the ankle), and the thigh (3 cm above the knee). The sampling rate of the sensors was set at 100 Hz [33]. In addition, the iSen values are the average of the maximum values for each movement within a single gait cycle. The anatomical position calibrated prior to the measurements is used to measure the angles of flexion and extension. The STT wireless wearable sensor system iSEN is a sensor modular system that communicates using Wi-Fi. The data transfer rate varies from 400 Hz to 25 Hz depending on the number of sensors. The iSEN sensors connect to the Wi-Fi common router as clients [34]. Studies have demonstrated that IMUs can have good to excellent test-retest reliability for measuring movement parameters such as gait speed, stride length, and joint angles. Studies have demonstrated good to excellent concurrent and criterion validity for IMU-based measures of gait speed, stride length, joint angles, and other movement parameters [35].

### 2.3. Experimental Protocol

In this study, participants were randomized into three groups to investigate the effects of visual feedback balance training on unstable surfaces, the effects of balance training on unstable surfaces, and the effects of conventional physical therapy. Before the experiment, participants were asked to familiarize their gait on a walkway without controlling their gait to induce natural motions.

The participants were randomly assigned to one of three groups: VUSBG, USBG, or CG according to the intervention method: (1) VUSBG: The visual feedback complex training group trained visual feedback in a sitting position with a laser pointer (Motion Guidance) attached to the center of the forehead, and a target with a diameter of 25 cm was attached at an eye level of 2 m to set a visual feedback target. Posture control training was conducted in a sitting position under the supervision of a therapist so as to prevent deviation of the green laser pointer from the target. Multi-Spine (WOORIMEDI, Medical Science, Republic of Korea; Model: MS-3000, Speed: 0~34 rpm, Angle: 0~13′, Weight: 210 kg, Dimensions: 1180 × 1900 × 2360 mm) equipment in the shape of an oval disk with a diameter of 120 cm was used to apply an unstable support surface (Figure 1). The disk was set to rotate at 6 m/s at an angle of 9° and the direction of rotation was reversed in minute increments. Proprioceptive and visual feedback stimulus interventions were applied as complex balance training in a sitting position. (2) USBG: The sitting balance training group used the same multi-Spine equipment in the shape of an oval disk with a diameter of 120 cm to apply an unstable support surface. The disc was set to rotate at 6 m/s at an angle of 9° and the direction of rotation was reversed in minute increments. The participant sat on the disc and trained for balance through posture control for 20 min. (3) The CG performed conventional rehabilitation treatment based on neuro development treatment such as mat exercises, muscle strength exercises, postural control exercises, and functional activity on a stable surface for 30 min, according to a physical therapy schedule. The VUSBG and USBG were performed 12 times a day for 20 min, 3 days a week, for 4 weeks. Depending on the participant’s exercise capacity, the training was stopped if they could not sustain it for 30 min, and a 5-min rest period was allowed if they showed fatigue, reported pain, demonstrated abnormal breathing, or appeared flushed [36]. Pre- and post-tests were performed prior to the experiment and at 4 weeks after the experiment, respectively (Figure 2). The evaluation examined the spatiotemporal and kinematic gait parameters (gait velocity, stride time, stride length, cadence, hip, and knee joint angle), balance ability (TIS, BBS), and gait ability (TUG).

### 2.4. Statistical Analysis

Descriptive data analyses and statistical tests were performed using SPSS version 25.0. (SPSS, Inc., Chicago, IL, USA). The normality of data was examined using the Shapiro–Wilk test. A paired t-test was used to investigate the effect of the intervention on balance and gait parameters. One-way ANOVA was used to analyze changes before and after the intervention for each group, and a post-hoc analysis with Bonferroni correction was performed to measure the magnitude of the changes (α = 0.017). Statistical significance was set at *p*-value < 0.05.

## 3. Results

### 3.1. Before-and-After Analysis of Each Group on General Characteristics

Table 1 shows the general characteristics of each group. No significant differences were observed in the demographic data between the groups in terms of age, sex, height, weight, K-MMSE, onset period, and affected hemisphere (*p* > 0.05).

### 3.2. Before-and-After Analysis of Each Group on Balance

In Table 2, VUSBG showed a significant increase in TIS and BBS scores after 4 weeks of the intervention (*p* < 0.05). USBG exhibited significant increases in the TIS and BBS scores after 4 weeks of the intervention (*p* < 0.05). No significant difference was observed in CG’s TIS and BBS parameters following the 4-week intervention (*p* > 0.05).

### 3.3. Before-and-After Analysis of Each Group on Gait Parameters

According to Table 2, both VUSBG and USBG interventions resulted in a statistically significant reduction in TUG values (*p* < 0.05), while CG intervention did not show a significant difference in TUG parameters after four weeks (*p* > 0.05).

In the VUSBG, the gait velocity and stride length significantly increased after 4 weeks of intervention (*p* < 0.05). However, cadence, stride time, and knee extension angle were not significantly different in the VUSBG compared to those in the USBG (*p* > 0.05). USBG showed significant increases in the gait velocity and stride length after 4 weeks of intervention (*p* < 0.05). However, cadence, stride time, and hip extension angle were not significantly different in the USBG compared to those in the VUSBG (*p* > 0.05). CG showed no significant difference in all spatiotemporal gait parameters according to the intervention period (*p* > 0.05).

VUSBG showed a significant increase in the hip flexion angle, hip extension angle, and knee flexion angle after 4 weeks of intervention (*p* < 0.05). However, the knee extension angle was not significantly different in the VUSBG compared to that in the USBG (*p* > 0.05). USBG showed significant increases in the hip flexion angle, knee flexion angle, and knee extension angle after 4 weeks of intervention (*p* < 0.05). However, the hip extension angle was not significantly different in the USBG compared to that in the VUSBG (*p* > 0.05). CG showed no significant difference in all kinematic gait parameters after 4 weeks of the intervention (*p* > 0.05).

### 3.4. Post-Hoc Analysis between Groups in Balance and Gait Parameters

As a result of post-hoc analysis of the change between each group, the TIS, TUG, and BBS scores were significantly higher in the VUSBG compared to those in the USBG and CG (*p* < 0.017). Regarding gait parameters, VUSBG had significantly greater gait velocity and stride length than CG (*p* < 0.017). In addition, hip flexion and knee flexion angles were significantly greater in the VUSBG than in other groups (*p* < 0.017). However, other parameters (cadence, stride time, hip extension angle, and knee extension angle) did not show significant differences between the groups (*p* > 0.017).

In addition, USBG showed significantly greater TIS and BBS scores than CG. Concerning the gait parameters, USBG had significantly greater gait velocity and stride length than CG (*p* < 0.017). In addition, hip flexion and knee flexion angles were significantly greater in the USBG than in the CG group (*p* < 0.017). However, other parameters (cadence, stride time, hip extension angle, and knee extension angle) did not show significant differences between the groups (*p* > 0.017).

## 4. Discussion

The purpose of the present study was to investigate the effects of balance exercises and visual feedback on complex exercises. These exercises were based on proprioceptive neuromuscular facilitation on unstable support surfaces. The aim was to understand their utility in promoting the recovery of balance and gait ability in chronic stroke patients. According to the findings of a before-and-after comparison, VUSBG showed significant improvement in functional balance score (TIS and BBS) and gait parameters, such as TUG, gait velocity, stride length, hip flexion/extension angle, and knee flexion angle over the intervention period, similar to the findings of a previous study [37]. This may be due to the fact that the stroke patients in VUSBG received intensive and repetitive balance training with visual feedback, allowing them to perform challenging trunk control tasks [38]. From the point of view of gait analysis, previous research suggests that visual feedback training with an unstable surface is accomplished by increased temporal parameters and stride, rather than by cadence [39]. The findings revealed that stride length increased gait velocity by increasing trunk stability through kinematic changes in the hip and knee joints; however, cadence and stride time were relatively unaffected.

USBG showed significant improvement in functional balance score and gait parameters such as gait velocity, stride length, hip flexion angle, and knee flexion angle after the intervention period. However, USBG showed no significant difference in cadence, stride time, and hip/knee extension angle. Previous research has found that 4 weeks of balance training improves patients’ balance and gait functions [40,41]. Training on an unstable surface induces reactive postural control in the trunk muscles for balance, improving muscle activity and trunk control more than training on a stable surface [42]. Although there were no significant differences observed in kinematic data at the hip joint, hip extension is important because it advances the trunk segment forward over the stance foot, which contributes to normal contralateral step length [43,44]. Weight shifting on an unstable surface enhanced selective movement and symmetrical postural control of the trunk in this study; however, it is considered an insufficient intervention method to alleviate kinematic disruption in stroke patients.

After the conventional intervention, there was no significant difference in the functional balance and gait characteristics in the CG. Previous studies designated the CG as the NDT group, and these studies did not reveal any significant differences in gait and balance despite intervention for at least 6 weeks [45,46]. These results are consistent with our study. A study that used NDT for more than 3 months, on the other hand, found significant improvements in gait and balance functions. The improvement was associated with a 3-month continuous application of the method. Implementation of a short 3-week neurorehabilitation program enhanced kinematic gait parameters but had no effect on the central programming step. Hence, a reassessment of existing programs was carried out to include neurorehabilitation daily, for at least 28 days during the kinesis therapeutic program [47]. We believe that the lack of significant change in CG’s functional balance and gait characteristics can be attributed to the relatively short intervention period.

Post-hoc analysis revealed that VUSBG had a significant main effect on functional evaluation (TIS and BBS) scores and gait parameters (TUG, hip flexion angle, and knee flexion angle) compared to the USBG. Among the gait parameters, hip flexion and knee flexion angles were significantly increased in the USBG compared to the CG. A head-mounted laser can provide visual feedback, which may help improve center of mass (COM) control after a stroke. This visual cue may have had a greater effect on the stroke group as a result of their increased reliance on visual cues for balance control [48]. Furthermore, the laser offered feedback on body movement, which might be used to compensate for poor trunk position perception [49]. This strategy is based on motor learning theories that are essential for the central nervous system to function properly, such as task-oriented plasticity caused by focused attention, repetition of desirable motions, and proprioceptive feedback [50]. All of these strategies are beneficial for enhancing gait recovery, although evidence shows that these strategies have modest reliability and reproducibility. In some studies, the training period was designed to be 4 weeks [51,52] and 5 weeks [53]. In this study, the participants trained for 4 weeks, and it was found that visual feedback self-exercise could improve postural control in a short duration of time. In terms of kinematic parameters, Nadeau et al. suggested that the hip flexor is more essential than other muscles in compensating for distal impairments on the paretic side [54,55]. If dorsiflexion is sufficient for toe clearance, a smaller amount of peak hip flexion is required; however, if dorsiflexion is insufficient, hip flexion is increased to reduce the risk of falls. Understanding patients’ strategies is crucial, as Chen et al. revealed. They found that selective proximal motor control of the lower limb, rather than distal control, could significantly affect final gait velocity [56]. Thus, this study was based on the assumption that trunk stability increased following exercise on an unstable support surface, enabling joint movement to compensate for the ankle joint restriction caused by stroke.

When comparing VUSBG and CG, post-hoc analysis revealed that VUSBG had a significant main effect on functional evaluation (TIS and BBS) scores and gait parameters (TUG, gait velocity, stride length, hip flexion angle, and knee flexion angle). The reason VUSBG showed a significant difference compared to CG could be due to improvement in postural stability by the task-oriented movements and proprioceptive feedback as described above, due to which the kinematic variables of gait were improved [39,41]. According to a recent systematic review [42], several studies conducted interventions on unstable surfaces for at least 4 to 8 weeks. The study conducted on unstable surfaces for 2 weeks has low reliability due to inadequate data on results, and studies with interventions for 4 weeks or more showed significant differences in balance and gait functions [57,58,59]. The present study is consistent with previous studies, which demonstrate that balance training on an unstable surface needs to be performed for at least 4 weeks, to be better than conventional intervention. However, the present study has several limitations. First, the lasting effect cannot be determined due to the short experimental period and lack of follow-up. Second, control over dual tasks and movements during the participants’ daily activities was insufficient. Third, the patient sample was not large enough to determine whether the findings could be generalized to other populations. Furthermore, due to the small sample size, the statistical results may be influenced, and therefore, it is necessary to increase the sample size in future research. Finally, we were unable to classify subtypes of stroke in our recruited participants. In addition, while it is true that individual differences may affect the outcomes of the study, the results are still valuable and informative for clinicians and researchers. The findings can be used as a basis for developing individualized rehabilitation programs that take into account the specific needs and responses of stroke patients. Further investigation is needed to determine whether visual feedback on an unstable support surface with balance exercise leads to motor relearning and neuroplasticity in chronic hemiparetic patients.

## 5. Conclusions

In conclusion, the present study was conducted to investigate the effects of balance exercises and visual feedback with complex exercises, based on the proprioceptive neuromuscular facilitation on unstable support surfaces, for recovery of balance and gait ability in chronic stroke patients. The results indicate that balance and gait ability improved significantly associated with unstable surface balance training based on proprioceptive neuromuscular facilitation (VUSBG and USBG). One potential implication is that visual feedback balance training improves balance and gait ability in chronic stroke patients. Visual feedback complex exercise based on the principle of proprioceptive neuromuscular facilitation could be an intervention strategy to improve gait speed, trunk stability, and mobility in chronic stroke patients.

## Figures and Tables

**Figure 1 jcm-12-04383-f001:**
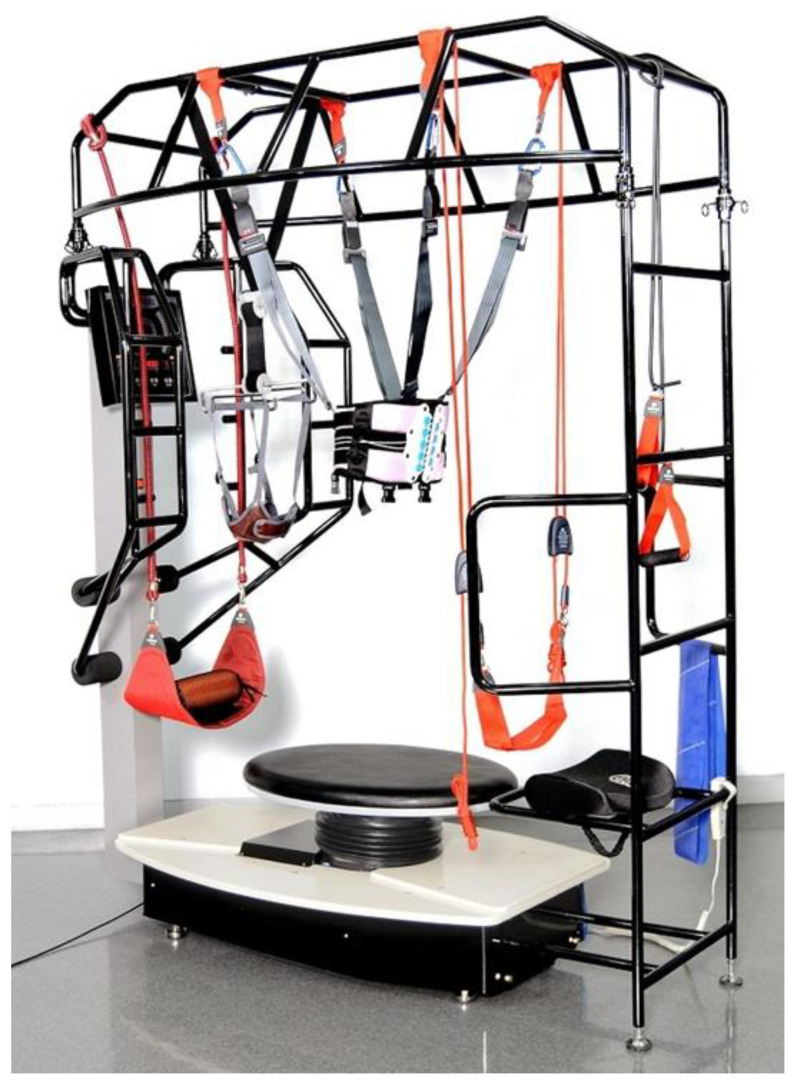
Multi-spine equipment.

**Figure 2 jcm-12-04383-f002:**
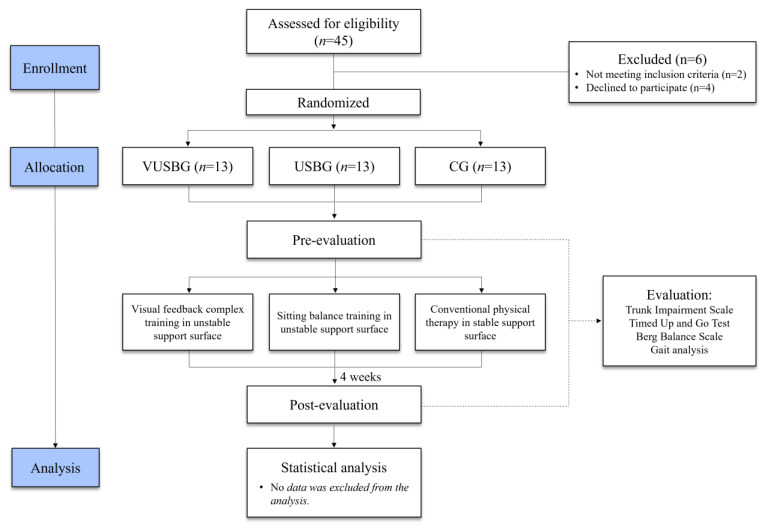
Flow chart of the experimental protocol.

**Table 1 jcm-12-04383-t001:** General characteristics of each group.

	VUSBG	USBG	CG
Age (years)	44.85 (15.63)	56.92 (8.95)	51.54 (12.74)
Sex (male/female)	9/4	11/2	11/2
Height (cm)	168.69 (8.84)	167.08 (8.39)	167.46 (10.83)
Weight (kg)	67.00 (7.30)	69.23 (8.12)	65.38 (4.41)
K-MMSE (score)	27.08 (1.85)	28.23 (1.24)	28.00 (1.83)
Onset period (month)	15.15 (4.67)	14.92 (5.42)	16.85 (4.74)
Affected hemisphere (Rt/Lt)	8/5	7/6	5/8
Subtype (Infarction/Hemorrhage)	8/5	8/5	11/2

Values represent mean (±standard deviation); VUSBG: Visual feedback combined with unstable surface balance training group; USBG: Unstable surface balance training group; CG: Conventional physical therapy group.

**Table 2 jcm-12-04383-t002:** Comparison of functional balance and gait assessment according to the intervention period for each group.

	Variables	Group	VUSBG	USBG	CG	F(p)	Post-Hoc Comparisons		
							A vs. B	A vs. C	B vs. C
Balance	TIS (score)	Baseline	13.00 (4.56)	13.23 (2.56)	15.15 (2.88)	118.255 (<0.001) *	<0.001	<0.001	0.003
		4 weeks	20.00 (4.26)	15.23 (2.56)	15.54 (2.82)				
		*p*	<0.001	<0.001	0.096				
	BBS (score)	Baseline	24.08 (6.21)	23.08 (5.84)	25.54 (6.86)	110.092 (<0.001) *	<0.001	<0.001	0.006
		4 weeks	32.31 (6.34)	25.08 (6.10)	25.62 (7.05)				
		*p*	<0.001	<0.001	0.584				
Gait ability	TUG (s)	Baseline	37.91 (16.59)	23.19 (13.04)	24.66 (12.12)	24.393 (<0.001) *	<0.001	<0.001	0.554
		4 weeks	21.53 (8.95)	19.16 (11.18)	23.82 (11.98)				
		*p*	<0.001	<0.001	0.350				
	Cadence (step/min)	Baseline	73.47 (17.00)	79.25 (24.06)	82.96 (14.93)	0.266 (0.768)	1.000	1.000	1.000
		4 weeks	76.00 (19.97)	82.88 (25.29)	83.85 (13.75)				
		*p*	0.391	0.281	0.615				
	Gait velocity (m/s)	Baseline	0.38 (0.20)	0.48 (0.26)	0.65 (0.39)	14.101 (<0.001) *	0.122	<0.001	0.010
		4 weeks	1.15 (0.60)	0.95 (0.40)	0.67 (0.31)				
		*p*	<0.001	<0.001	0.767				
	Stride time (s)	Baseline	1.71 (0.65)	1.75 (0.71)	1.52 (0.38)	0.280 (0.758)	1.000	1.000	1.000
		4 weeks	1.61 (0.56)	1.63 (0.57)	1.50 (0.38)				
		*p*	0.394	0.414	0.502				
	Stride length (m)	Baseline	0.59 (0.23)	0.71 (0.26)	0.94 (0.51)	13.356 (<0.001) *	0.179	<0.001	0.009
		4 weeks	1.75 (0.76)	1.45 (0.67)	0.96 (0.33)				
		*p*	<0.001	0.002	0.854				
	Hip flexion (°)	Baseline	12.00 (6.89)	15.51 (10.40)	24.63 (12.56)	45.908 (<0.001) *	0.001	<0.001	<0.001
		4 weeks	30.24 (11.73)	26.05 (11.63)	24.41 (11.82)				
		*p*	<0.001	<0.001	0.588				
	Hip extension (°)	Baseline	2.15 (3.78)	3.51 (3.82)	5.99 (7.30)	4.267 (0.022) *	0.533	0.018	0.394
		4 weeks	4.00 (4.06)	4.42 (4.50)	5.84 (7.15)				
		*p*	0.017	0.078	0.418				
	Knee flexion (°)	Baseline	26.77 (10.23)	27.31 (9.76)	37.43 (18.93)	70.958 (<0.001) *	<0.001	<0.001	<0.001
		4 weeks	50.13 (10.70)	40.52 (11.24)	37.51 (18.97)				
		*p*	<0.001	<0.001	0.530				
	Knee extension (°)	Baseline	2.23 (3.10)	2.42 (2.36)	2.09 (3.26)	0.456 (0.638)	1.000	1.000	1.000
		4 weeks	2.69 (3.33)	2.94 (2.71)	2.05 (3.06)				
		*p*	0.508	0.181	0.802				

Values represent mean (±standard deviation); VUSBG(A): Visual feedback combined with unstable surface balance training group; USBG(B): Unstable surface balance training group; CG(C): Conventional physical therapy control group; TIS: Trunk Impairment Scale; TUG: Timed Up and Go; BBS: Berg Balance Scale; * significant difference between the groups (*p* < 0.05).

## Data Availability

The data presented in this study are available on request from the corresponding author. The data are not publicly available due to personal data protection.

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
