# Peer review of "Effect of Balance Training in Sitting Position Using Visual Feedback on Balance and Gait Ability in Chronic Stroke Patients"

_jcm, 2023, doi:10.3390/jcm12134383_

Round 1
Reviewer 1 Report (Previous Reviewer 1)
The authors investigated the effects of visual feedback with balance exercises and complex exercises based on proprioceptive neuromuscular facilitation on unstable support surfaces for restoring balance and walking ability in chronic stroke patients. The results show that balance and walking ability are significantly improved in association with unstable surface balance training based on proprioceptive neuromuscular facilitation.
Some Comments:
A paragraph should be added at the end of the first section on what to do in the other sections of the article.
The article should be revised grammatically.
For the statistical analysis part, parametric tests were performed according to the normality of the data. In this section, attention is paid to the following.
Paired t test is used when there are 2 dependent groups. ANOVA is used when there are 3 or more independent groups. If groups are dependent and there are 3 or more groups, repeated measures analysis of variance should be used. Authors should pay attention to this situation.
There is no conclusion section in the article. A new section should be opened and the results of the article should be clearly stated.
Minor editing of English language required
Author Response
Thank you for your thorough review. It has been very helpful in improving our research.
Please see the attachment.

Reviewer 2 Report (New Reviewer)
Dear authors,
congratulations for your work about the Effect of Balance Training in Sitting Position Using Visual Feedback on Balance and Gait Ability in Chronic Stroke Patients. You aimed to see the combined effect of unstable surface balance training and visual feedback, based on proprioceptive neuromuscular stimulation in patients with chronic stroke, was effective in restoring balance and gait ability. In general the article is well written and the language used is clear for the reader. I have a few improve suggestions that are add more recent literature in discussion and introduction since you have less than 10 articles from the last 5 years and you should improve your tables to tried to reduce them.
Abstract: Add sentence in the beginning about your topic; Add data in results section;
Introduction: Add more recent studies. I suggest to move the following sentence : “Previous research has demonstrated that exercise interventions can enhance balance and gait in stroke patients. However, studies that compare the effectiveness of combined exercise interventions are limited” to the beginning of the paragraph. E.g.: “Previous research has demonstrated that exercise interventions can enhance balance and gait in stroke patients. However, studies that compare the effectiveness of combined exercise interventions are limited. Therefore the purpose of this study is to investigate the effect of exercise that combines visual feedback and balance training on the balance and gait of patients with chronic stroke. Balance and gait impairments are prevalent among stroke patients, and these impairments have a substantial impact on their quality of life. The independent variables of this study consisted of types of exercise interventions, which included visual feedback and an unstable support surface. The dependent variables for this study will encompass the measurement of balance and gait through the use of standardized assessments.”
Methods: Are clear and precise to be replicated.
Results: Improve table 1 and 2
Discussion: Add more recent studies.
Conclusion: Add conclusion section;
Language used is clear and understandable for the reader.
Author Response
Thank you for your thorough review. It has been very helpful in improving our research.
Please see the attachment.

Round 2
Reviewer 1 Report (Previous Reviewer 1)
The specified corrections have been made by the authors in detail and the quality of the article has been increased. The article is acceptable as it is.
This manuscript is a resubmission of an earlier submission. The following is a list of the peer review reports and author responses from that submission.
Round 1
Reviewer 1 Report
The contents and structure of the paper is appropriate. However, I have some minor comments to improve the paper:
Update the references of your manuscript to include at least 5 references published from 2020 to 2023
Extend abstract, introduction, and conclusion in comprehensive manner.
The originality of the paper needs to be further clarified.
A paragraph describing what was done in the other chapters can be added at the end of the introduction.
References should be cross checked from text and reference list and some relevant references from interest journa may be added.
When measurements are taken from the same individuals at different times, such data are dependent (paired) data. If the data is normal in the comparison of 2 dependent groups, the Paired samples t test is used, and if the data is not normal, the Wilcoxon test is used. When comparing more than 2 dependent groups in terms of means, it uses the Repeated Measures analysis if the data is normal, and the Friedman test if the data is not normal.There is a before and after in the related article, which indicates that these groups are dependent. However, ANOVA test was used to compare more than 2 groups. ANOVA compares independent groups. This part should be re-evaluated by the authors.
The contents and structure of the paper is appropriate. However, I have some minor comments to improve the paper:
Update the references of your manuscript to include at least 5 references published from 2020 to 2023
Extend abstract, introduction, and conclusion in comprehensive manner.
The originality of the paper needs to be further clarified.
A paragraph describing what was done in the other chapters can be added at the end of the introduction.
References should be cross checked from text and reference list and some relevant references from interest journa may be added.
When measurements are taken from the same individuals at different times, such data are dependent (paired) data. If the data is normal in the comparison of 2 dependent groups, the Paired samples t test is used, and if the data is not normal, the Wilcoxon test is used. When comparing more than 2 dependent groups in terms of means, it uses the Repeated Measures analysis if the data is normal, and the Friedman test if the data is not normal.There is a before and after in the related article, which indicates that these groups are dependent. However, ANOVA test was used to compare more than 2 groups. ANOVA compares independent groups. This part should be re-evaluated by the authors.